# Biphasic Porous Bijel-Like Structures with Hydrogel Domains as Controlled Drug Delivery Systems

**DOI:** 10.3390/gels10010072

**Published:** 2024-01-18

**Authors:** Fabio Pizzetti, Giovanna Massobrio, Silvia Riva, Francesco Briatico Vangosa, Filippo Rossi

**Affiliations:** Department of Chemistry, Materials and Chemical Engineering “Giulio Natta”, Politecnico di Milano, Piazza Leonardo da Vinci 32, 20133 Milan, Italy; giovanna1.massobrio@mail.polimi.it (G.M.); silvia2.riva@mail.polimi.it (S.R.); francesco.briatico@polimi.it (F.B.V.)

**Keywords:** alginate, bijels, drug delivery, polymer, rheology

## Abstract

Bijels are a peculiar type of Pickering emulsion that have a bicontinuous morphology and are stabilised by a jammed layer of nanoparticles (NPs). Due to their double nature, their usage has increased in recent years in various fields, such as biological and food applications. In fact, they can release both hydrophilic and hydrophobic compounds simultaneously. An improvement to this structure is the use of a hydrophobic monomer like polycaprolactone as the organic phase, which is able to polymerise during the formation of the structure. Unfortunately, the structures formed in this way always have some drawbacks, such as their thermal stability or degradation when submerged in an aqueous medium. A number of studies have been carried out in which some parameters, such as the NPs or the monomer, were changed and their effect on the final product evaluated. In this work, the effect of modifying the aqueous phase was studied. In particular, the effect of adding alginate, a biopolymer capable of forming a stable hydrogel in the presence of divalent cations, was analysed, as was the difference between soaking or not in CaCl_2_, the final system. Specific attention was paid to their swelling behaviour (150% vs. 25% of the blank sample), rheological properties (G′ 100 kPa vs. 20 kPa of the blank sample) and their release performances. In this framework, complete release of hydrophilic drug vs. 20% in the blank sample was observed together with improved release of the hydrophobic one with 35% in 8 h vs. 5% in the case of the blank sample. This strategy has been proven to influence bijels’ properties, opening the doors to many different uses

## 1. Introduction

Bicontinuous interfacially jammed emulsion gels, also known as bijels, are a particular type of Pickering emulsion in which the aqueous and organic phases are present in almost equal volume ratios [1]. They are stabilised using nanoparticles (NPs) instead of classical surfactants, which also increases their biocompatibility and lowers their toxicity. NPs in this configuration are able to position themselves at the interface between the two phases, forming a jammed monolayer capable of stabilising such a conformation [2,3]. Unlike classical Pickering emulsions, bijels have a bicontinuous domain due to the ratio between the different phases and the concentration of NPs [4,5]. These peculiar structures are becoming increasingly important in different fields, such as catalysis [6,7], food [8], molecule encapsulation [9,10], tissue engineering [4,11] and drug delivery [12,13,14]. The latter derives from their double nature, which enables them to simultaneously load and release both hydrophilic and hydrophobic drug mimetics, thus enhancing their usage as drug delivery systems (DDSs) [15]. In general, the most common materials available in recent years as DDSs have been hydrogels and nanogels, mainly thanks to their biocompatibility, biodegradability and tunability to achieve a targeted DDS [16,17,18,19]. 

A disadvantage of these classical systems is their high hydrophilicity, which limits their use for hydrophobic drugs. Classical bijels are produced using two different liquids, one aqueous and one organic, and the final structure can be compared to a foam [4,20]. Several modifications can be applied to make these systems more suitable for certain applications. One of these is the substitution of the classical oil phase with a monomer that is able to polymerise during structure formation. Several studies have been carried out in this direction to evaluate any changes in the final material properties and structures, such as thermal stability and pore size [20,21,22]. Several parameters play a relevant role in this tuning, with the choice of monomer and the size and type of nanoparticle probably being the most important. A number of studies have been carried out on these two parameters in order to assess their impact on the final structure. Based on the monomer study, initial analyses were carried out using monomers differing only in the number of carbon atoms. Partenope et al. [23] shifted to ω-pentadecalactone, starting from the classical formulations produced using ε-caprolactone (CL) [15]. This change improved the thermal stability and also decreased the pore size of the materials, thus changing their release performance. Unfortunately, these advantages came with some disadvantages, in particular the higher temperatures required for synthesis, which could eventually damage molecules that need to be loaded into it. The different types of NPs, in particular the transition between inorganic and organic NPs, have also been studied. Vanoli et al. [6] characterised the release properties of CL-based bijel-like structures produced using inorganic and organic nanoparticles. Some effects have also been observed with this change, producing different release profiles for both hydrophilic and hydrophobic drugs. Unfortunately, the drawback of the high temperature needed for the synthesis to change the monomer could be an overwhelming limitation if thermosensitive molecules were to be used. Furthermore, several numerical studies have also been carried out on the effect of the NPs’ dimensions and concentration in the aqueous medium. An example of this work is that of Jansen et al. [20], which studied the effect of NPs on the transition from a classical Pickering emulsion and a bijel. One aspect that requires further investigation is the modification of the aqueous phase. There are two different ways of doing this: changing the aqueous phase with a different solvent or using hydrophilic additives. As regards the former, Bai et al. [24] tried producing bijels using two different non-polar solvents. The problem associated with this approach for biological applications is finding two different biocompatible and biodegradable non-polar solvents and finding one that is capable of complete polymerisation. The latter is probably the most viable solution for biomedical applications. In this work, a new component to be added in the aqueous phase was studied: alginic acid sodium salt (also referred to as alginate). Alginate is a copolymer with two different repeating units of mannuronic and guluronic acid, capable of forming a gel when placed in a solution containing a bivalent cation such as Ca^2+^ [25,26]. Since the final application is likely to be something that requires the biocompatibility and biodegradability of the material, alginate was also selected because it possesses these properties and is widely used in a variety of applications ranging from drug delivery to the food industry [27,28,29]. As an example, Takatsuka et al. [30] developed an alginate-based hydrogel microbead for gene delivery via adeno-associated viruses (AAV). In this study, different concentrations of alginate in the aqueous phase were analysed, in particular their effect on rheological and release properties. Alginate is a water-soluble molecule, but its concentration has a strong influence on the viscosity of the final solution. This, coupled with the divalent cation concentration and its soaking time, allows the development of hydrogels with different rheological behaviours. Several studies were carried out focusing on the properties of divalent cation solutions. In particular, different soaking times in a CaCl_2_ solution were tested. This was also necessary to assess the time taken for calcium ions to diffuse into the structure and form hydrogel networks. Even though every divalent cation is able to produce a hydrogel, only Ca^2+^ was tested in this study because its better stability has been reported in several works [31]. 

Morphological analyses, especially scanning electron microscopy (SEM), were performed to assess any changes in the structure of the devices due to the variations in the alginate concentration and the soaking time. Thermal analyses were also carried out to assess any change in the melting temperature caused by changes in the formulations. Furthermore, due to the presence of a new component in the aqueous phase, swelling tests were also carried out and compared with the response of blank samples produced without alginate. This analysis focused on rheology and release studies to investigate the effect of the additive (alginate) used in this novel formulation. Rheology was considered to be of paramount importance because, as previously mentioned, one of the major disadvantages of bijels is their handling. Rheological measurements were performed in both amplitude sweeps and frequency sweeps. Amplitude sweeps were carried out to assess the extension of the linear viscoelastic region. Frequency sweeps were then performed to investigate the viscoelastic behaviour and the effect of the addition of alginate and cross-linking. The storage modulus G′ value was taken as an indication of the stiffness of the bijel and was also used to assess which samples had the best solid-like behaviour. Turning to release assays, since a potential final application of the bijel is a DDS, various drug mimetics were selected to assess their release properties. Both hydrophilic and hydrophobic were selected, and the drug delivery performances were compared. 

## 2. Results and Discussion

### 2.1. Biphasic Porous Structure Synthesis and Characterisation

Bijel-like structures were produced according to the procedure presented in the Materials and Methods section and visible in Figure 1.

Briefly, during step (a) ε-caprolactone, TBD and EtOH are added to a syringe to start the polymerisation. In step (b), the mixing is stopped, and the aqueous solution of NPs and alginate is added to the system. Mixing is restarted until bijel formation in step (c). The bijel-like structure is then soaked in a CaCl_2_ solution, allowing the formation of a hydrogel in the bijel’s aqueous domain (step (d)). Initially, some characterisation tests on the structures produced were performed. Based on the thermal properties, DSC analyses were carried out to assess the thermal stability of the samples, and the melting and crystallisation temperatures, in particular, were evaluated. Attention is drawn to the change in the value of this temperature simply by adding different concentrations of alginate in the aqueous phase and changing the soaking time in the CaCl_2_ solution. The results of such analyses are reported in Figure 1. 

The structure obtained without alginate (blank) thermogram is shown in Figure 1 for comparison, while the role of alginate concentrations soaked in CaCl_2_ solution can be observed in Appendix A. In this comparison, no differences in melting temperature were observed between the samples. Some differences can be seen when comparing a soaked and an unsoaked bijel. In fact, the addition of alginate in polymer form to the bijel formulation increases the melting temperature compared to the bijel after being soaked in CaCl_2_. In that condition, alginate is present in gelled polymer form, and its melting point is similar to the one of the blank sample (without alginate). After soaking, the sample is placed in the aqueous CaCl_2_ solution, and the melting temperature reverts to the value of the blank bijel (Figure 1). It can be stated that the addition of alginate to the aqueous solution results in an increase in the structural thermal properties of the unsoaked bijel. When looking at the immersion in an external aqueous solution, the thermal properties tend to decrease, but with the addition of alginate after 3 h, the same behaviour of the classical unsoaked bijel can be observed. The effect of the addition was also seen in the morphology of the final structures, analysed through scanning electron microscopy. The recorded images are shown in Figure 2. From what can be observed, the final structure strongly resembles that of the classical bijels, already reported in previous works by our research group [6,15]. Some new information can also be obtained. For example, the SEM images are reported for unsoaked samples and samples soaked in CaCl_2_ solution, and those after three hours show more lumps compared to the unsoaked or classical bijel (Appendix A). EDS analysis revealed that these lumps were simply deposits of calcium chloride (Appendix A).

Other than this difference, no major variation in the morphology can be observed between the unsoaked and soaked samples, thus ensuring that the resting in CaCl_2_ solution does not affect the matrix. When comparing the different alginate concentrations, no major differences were observed, confirming that increasing the alginate concentration does not affect the matrix architecture (Appendix A).

Additional tests involved the evaluation of the swelling behaviour. Trends in swelling kinetics are shown in Figure 3. Unlike classical hydrogels, bijels generally do not have a high swelling ratio at equilibrium due to their stiffer structures and high hydrophobic moiety content. The difference here is that the aqueous phase consists of a dissolved polymer that could form a hydrogel after soaking in a solution containing divalent cations. The system is observed to reach equilibrium after the first few minutes, and as can be deduced, the 20 mg/mL samples showed a higher equilibrium swelling ratio (Appendix A). Furthermore, when compared with the blank samples (without alginate), it can be seen that the equilibrium swelling ratio is one order of magnitude higher. It is clear that adding a gelling agent to the hydrophilic domains of biphasic porous structures increases the ability to retain water within the 3D network.

### 2.2. Rheological Measurements

Rheological measurements were carried out to assess the rheological behaviour of the final structure of the different samples.

Initial rheological measurements focused on amplitude sweep tests to assess the extension of the linear viscoelastic region (LVR). The results of the amplitude sweep tests for bijels produced with an alginate concentration of 10 mg/mL are shown in Figure 4. Based on the LVR, neither the presence of alginate nor the soaking time seem to affect its range, as the linear region is similar to that of the blanks. What appears to be affected by the soaking time is the value of G′ in the linear range. Furthermore, in the medium-amplitude oscillatory shear (MAOS) and large-amplitude oscillatory shear (LAOS) regions, the slope of G′ differs slightly between blank and alginate samples. At the larger shear strain amplitude, the alginate containing biphasic porous structures shows the same slope regardless of the soaking procedure.

As regards frequency sweep tests, comparisons were made between the change in the storage (G′) and loss (G″) moduli. Following this, an analysis was performed on the different values of G′, comparing only the different soaking times at different alginate concentrations. Initial analyses were performed on two alginate concentrations, namely 10 mg/mL (Figure 5) and 20 mg/mL (Appendix A). Starting with the former, initial frequency sweep tests showed no crossover between G′ and G″, always indicating a solid-like behaviour of the sample. Instead, by focusing on the G′ value for the different soaking times, some important information can be obtained (Figure 5).

First, the highest G′ is shown for the unsoaked sample with alginate. This means that the addition of alginate results in an increase in the solid-like behaviour of the porous structure. The other sample with a high G′ value is the blank sample, whereas all the soaked bijels have lower G′ values due to the presence of gel instead of solid polymers. The implications are that 10 mg/mL alginate is probably too low a concentration to see any improvement in material properties and that the effect of adding water to the sample outweighs the improvement due to the formation of hydrogel. To better assess this, analyses were also performed on 20 mg/mL samples. In theory, an increase in the concentration of alginate should lead to an increase in the G′ value, even after soaking in CaCl_2_ solution.

Based on the analysis of the comparison between G′ and G″, a crossover can be observed starting from the samples soaked for 30 min, and such a crossover tends to be present at lower frequencies as the soaking time is increased, probably due to the high amount of water present or the degradation of the polymer backbone. The comparisons of the G′ values for all the samples at 20 mg/mL are shown in Appendix A. Surprisingly, the G′ of the unsoaked bijel is lower than the blank one. These results could possibly be justified by the assertion that an increase in alginate in the aqueous phase even starts to introduce some steric hindrance into the matrix. This is likely to be reflected in the fact that the unsoaked bijel is softer compared to the blank. After 30 min of soaking, the G′ of the unsoaked sample still remains the same. After 3 h, however, it increases and reaches the value of the blank bijel. Up to this point, it could be deduced that this time is needed to allow the diffusion of CaCl_2_ inside the matrix, thus forming the hydrogel network. What might also seem strange at first is the further decrease in the G′ value after 24 h, which is even lower than the unsoaked value. In reality, this is similar to what was observed for the 10 mg/mL samples, i.e., the addition of water to the structure tends to make it softer. Two different effects are probably involved here: one related to the strengthening of the structure due to the hydrogel formation and an opposite one related to the decomposition of the polycaprolactone backbone due to water immersion. 

In the case of 10 mg/mL, the alginate concentration is so low that the second effect always prevails, i.e., the G′ value of the soaked gel is always lower. With regard to the sample containing 20 mg/mL alginate, we have two different domains. Initially, the first effect prevails with a higher G′ value, but then structural degradation takes place, resulting in a softer structure. In order to better assess this, two further tests were carried out, one with a concentration of 15 mg/mL to check whether G′ is somewhere between the two cases tested and one with 30 mg/mL to verify whether the first effect can become predominant. The results of these two tests are shown in Appendix A. For the 15 mg/mL test, the results are as predicted. As the alginate concentration is probably still limited, the G′ of the bijel after soaking decreases. Furthermore, the blank samples also showed a higher G′ value, supporting the hypothesis that as the alginate concentration increases, so does the steric hindrance, thus causing a decrease in the stiffness of the structure. Looking at the 30 mg/mL test, the results seem to confirm the initial hypotheses. Comparing the initial G′ of the blank and the unsoaked sample, the same results as those in the previous cases can be observed, i.e., the blank sample has a higher G′ value. With regard to the different soaking times, the results are consistent with the hypothesis made. More specifically, the 30 min and 3 h samples have similar G′ values. However, they are lower than those of the unsoaked bijel. As before, this is due to the time needed for the calcium ions to reach the inner part of the matrix and initiate hydrogel formation. In the sample soaked for 24 h, an increase in the G′ value can be observed. This means that the hydrogel has formed and dominates over the degradation of the polymeric backbone (PCL) due to the addition of water. Another surprising result is that the G′ value of the 24 h sample is even higher than that of the blank, indicating better properties. It can be concluded that the alginate can improve the rheological properties of the bijel, but its concentration and soaking time in the CaCl_2_ solution play a pivotal role.

### 2.3. Release Tests

Release tests are essential to assess the performance of all tested molecules in the final cumulative release. Thanks to its dual nature, this biphasic porous structure can indeed be useful for loading and releasing both hydrophilic and hydrophobic drugs. Several drugs have been studied in previous works [6,15,23], and the same molecules were selected for comparisons with blank bijel samples. Three different drug mimetics were chosen, namely fluorescein and rhodamine B for the aqueous phase and FITC for the organic phase. The releases obtained are compared with the results for blank bijels already published in previous works [15].

As can be seen in Figure 6, the presence of alginate has a major influence on the release of fluorescein, as in both cases the amount released is up to 100% after approximately 50 h. These results are completely different from the release of the blank bijels, which reaches 40% after 50 h. 

As well as a variation in the pore size of the matrix, this difference may also be due to some sort of interaction between the drug and the polymer added to the aqueous solution. In fact, the alginate polymer is present as the sodium salt of alginic acid and therefore has negative free charges. The same can be emphasised in the case of fluorescein, with the addition of a certain amount of electrostatic repulsion, which ultimately accelerates the final release of the drug. Figure 6b also shows the cumulative release against the square root of time, ensuring that the diffusion is Fickian for most of the release process before reaching the final plateau values. To better assess this aspect, RhB was used because of the difference in molecular charge. Unlike fluorescein, RhB has a relatively neutral charge at this pH and has approximately the same steric hindrance as fluorescein, which allows the charge effects alone to be studied.

With RhB, the results are different from those obtained using fluorescein (Figure 7). First, there is always an increase in the release profile, but not to the same extent as shown for fluorescein release. This could highlight the fact that some electrostatic interactions are also present, as discussed previously. 

When analysing Figure 7b, it can be seen that the release follows an initial Fickian regime, but the alginate samples showed a similar equilibrium value as the blank samples, unlike fluorescein. This confirms the observation that the alginate has little effect on the release kinetics.

FITC release tests were also carried out, demonstrating the ability of these structures to load and release even hydrophobic drugs as a multiple drug delivery system. This is because hydrophobic drugs must be loaded into the final structures by dissolving them in the organic solution prior to polymerisation. It has been observed in previous works that FITC can cause problems during the ROP of CL, probably due to interaction with TBD. The results of the release tests are shown in Figure 8. The effect of the higher alginate concentration is shown for completeness but should not be taken into account (Appendix A). 

Comparing the 10 mg/mL sample and the blank, the addition of alginate leads to an increase in the release profile for these molecules, as in the previous cases. It should also be highlighted that hydrophobic drugs are always released more slowly than hydrophilic drugs. This is due to their better affinity with the organic phase compared to the external aqueous environment. The higher profile in this case may also be due to this softening rather than the presence of the alginate itself. Figure 8b also shows the plot of cumulative release against the square root of time. The blank samples always show a Fickian diffusion regime, whereas for the alginate samples, it can be highlighted that the diffusion is Fickian before the degradation process starts.

## 3. Conclusions

Alginate is a natural polymer, able to produce hydrogels in the presence of divalent cations. The idea behind this work was to test whether the generation of a hydrogel matrix could improve the stability and rheological properties of biphasic porous bijel-like structures. Having determined that no effect on thermal properties could be detected by DSC analysis, attention turned to rheological and release properties. Based on the different concentrations of alginate tested, it was found that at low concentrations alginate causes an increase in the G′ value of the bijel, an effect that can be associated with a stiffening of the material. However, when the alginate concentration is increased beyond a certain value, the opposite effect is observed, probably due to the steric hindrance of the polymer. The effect of hydrogel formation was also investigated by soaking the bijel in CaCl_2_ solution for different periods of time. In this case, two different contributions must be taken into account. On the one hand, there is a stiffening effect due to the formation of the hydrogel, and on the other hand, the effect of water absorption could also introduce some negative forces that will cause the structure to collapse. In this case, different alginate concentrations and soaking times were used in several tests to check the effects on the final structures. It was observed that as the alginate concentration increases so does the diffusion time for CaCl_2_ inside the porous structure, showing that the time necessary to observe an increase in the G′ value increases too. At the highest concentrations tested, the bijel soaked for 24 h in CaCl_2_ solution had a higher G′ than the blank sample. It was observed that an increase in the alginate concentration leads to an increase in the stiffness of the bijel, but this requires a longer soaking time. Furthermore, for intermediate concentrations, after a certain time, the negative effect prevails, and G′ decreases to a lesser extent compared to the untreated one. In terms of release properties, an increase in the release profile was observed for all the drug mimetics tested (both hydrophilic and hydrophobic). As the alginate is in the form of alginic acid sodium salt, it has negative charges on its surface. This enhances the release kinetics due to electrostatic repulsion with negative molecule (here fluorescein) release while having almost no effect on the release rates of RhB and FITC. It can therefore be concluded that the addition of alginate could be a solution for enhancing the bijel properties, taking into account the key roles of polymer concentration and soaking time.

## 4. Materials and Methods

### 4.1. Materials

Fluorescein isothiocyanate isomer I (FITC, MW = 389.38 Da, ≥90%), fluorescein isothiocyanate-dextran 70,000 (FITC-DXT 70, MW = 70 kDa), fluorescein sodium salt (FLUOR, MW = 332.31 Da) and rhodamine B (RhB, MW = 479.01 Da) were purchased from Merck (Deisenhofen, Germany). Calcium hydroxide (Ca(OH)_2_, ≥98%) and orthophosphoric acid (H_3_PO_4_, ≥85%) were purchased from Carlo Erba Reagents (Carlo Erba Reagents, Milan, Italy). ε-caprolactone (CL, ≥99%), alginic acid sodium salt powder, calcium chloride and 1,5,7-triazabicyclo [4.4.0]dec-5-ene (TBD, ≥95%) were purchased from Sigma (Sigma-Aldrich Chemie GmbH, Deisenhofen, Germany). All reagents and solvents were used without further purification. The reactions were carried out in atmospheric air and synthesised products were stored at 4 °C in the dark until they were used.

### 4.2. Hydroxyapatite Nanoparticles Synthesis

Hydroxyapatite (HA) was synthesised according to the procedure reported in other works [6,15,23]. In summary, a 0.2 M solution of Ca(OH)_2_ (368.8 mg, 4.98 mmol, 10 eq.) in distilled water (24.9 mL) was heated up to 100 °C for 1 h with stirring. A 0.12 M solution of H3PO4 (0.2 mL, 2.99 mmol, 6 eq.) in distilled water (24.9 mL) was then added at a controlled rate of 4 mL/min. After this addition, the pH was adjusted to neutral, and the mixture was left to react for 2 h.

After overnight resting, the solution was centrifuged at 3000 rpm to recover the particles. The product was then dried at 200 °C for 6 h and ground in stages. The solution was stabilised by coating the NP surface with gum arabic (GA). A solution of GA was heated at 37 °C for 1 h with stirring, and then, a known mass of the NPs was added. The mass of NPs was chosen so that the final concentration value was equal to 20 mg/mL. After the addition, the dispersion was sonicated for 12 h. To produce the structure also containing alginate, alginate was added at a known concentration after sonication until complete dissolution.

### 4.3. Bijel-like Structure Synthesis

The biphasic porous structure synthesis has already been reported in other works from our research group [6,15]. In summary, ε-caprolactone (CL, 0.4 mL, 3.61 mmol, 1 eq.), ethanol (44 μL, 0.75 mmol, 0.2 eq.) and TBD (13 mg, 0.09 mmol, 0.025 eq.) were added to a double-opened syringe. The solution was mixed on a Heidolph Multi Reax shaker (Heidolph Instruments, Schwabach, Germany)equipped with a 12-rack carousel at 1000 rpm for 5 to 8 min at 16 °C. Following this, the same volume of the oil phase (400 μL) of aqueous dispersion of NPs was added to the reacting mixture, and the stirring speed was increased to 1700 rpm or 1900 rpm for 40 s. The former was selected for the blank samples and the latter for the structures also containing alginate. The stirring speed was then reduced to 1000 rpm until the structures were formed. The samples used for the release assays were always prepared following this procedure, but the drug mimetic was dissolved in the appropriate solvent (CL for FITC, aqueous dispersion of NPs for fluorescein and RhB). For the soaked samples, the produced devices were soaked for different periods of time in an aqueous solution of CaCl_2_ 2% *w*/*v*. The soaking times are specified in the Section 2.

### 4.4. Scanning Electron Microscopy (SEM)

SEM analyses were performed on gold-sputtered samples at 10 kV using an Evo 50 EP instrument (Zeiss, Jena, Germany). Samples were freeze-dried prior to analysis. Freezing was carried out by placing the samples in a −80 °C chamber for 2 h to ensure that the bijel matrix was not disturbed by the formation of too large ice crystals.

### 4.5. Differential Scanning Calorimetry (DSC)

DSC analyses were performed on a Mettler Toledo DSC Polymer machine (Mettler Toledo, Greifensee, Switzerland) calibrated with indium and zinc standards. The selected heating rate was 20 °C/min under nitrogen flow. The temperature varied between −60 °C and 180 °C, with two heating and cooling cycles. The sample was freeze-dried prior to analysis to avoid any peaks due to the freezing and evaporation of water.

### 4.6. Rheological Measurements

Rheological measurements were performed on an Anton Paar MCR 502 rheometer (Anton Paar, Graz, Austria) equipped with a 25 mm diameter stainless steel parallel plate configuration. The temperature was kept constant at 25 °C for every measurement. Both amplitude sweep and frequency sweep analyses were carried out. Amplitude sweep tests were necessary to evaluate the linear viscoelastic region (LVR), in particular, the upper limit. To do this, the shear strain amplitude varied between 0.001 and 10% at a fixed frequency of 1 rad/s. The limit of the LVR was obtained as the point after which G′ starts to show a decreasing trend as a function of shear strain amplitude. For the subsequent tests, the shear strain amplitude was fixed at 0.015%, i.e., within the LVR. Frequency sweep tests were then performed by varying the angular frequency of the oscillation between 0.1% and 100 rad/s.

### 4.7. Swelling Tests

Swelling measurements were carried out to assess any change in the swelling behaviour of the blank bijels. For this analysis, the samples were freeze-dried before testing, after having been frozen at −80 °C to maintain the internal structure.

After lyophilisation, the samples were weighed and then placed in water to check the increase in mass. Equation (1) was used to evaluate the swelling ratio, where *m*_0_ is the dried mass and *m_t_* the mass at time *t*.
(1)$$Q_{s}=\frac{m_{t}-m_{0}}{m_{0}}$$

### 4.8. Release Tests

Release assays were performed using different drug mimetics but only in the single release case study. For this assay, the drug mimetics were added during bijel formation by dissolving them in CL (a hydrophobic drug) or in the aqueous solution of NPs (hydrophilic drugs). The release was carried out in a controlled atmosphere and at 37 °C using an incubator to simulate physiological conditions. PBS was used as the release medium to maintain the pH at 7.4. The biphasic porous structure was therefore placed in 2 mL of PBS, and withdrawals were carried out at fixed time intervals, with the addition of fresh PBS to maintain the concentration gradient. Withdrawn samples were analysed spectrophotometrically at the wavelength of the absorption peak for each drug mimetic and analysed by the Lambert–Beer method.

## Data Availability

The data presented in this study are openly available in article.

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
