# Peer review of "Biphasic Porous Bijel-Like Structures with Hydrogel Domains as Controlled Drug Delivery Systems"

_gels, 2024, doi:10.3390/gels10010072_

Round 1
Reviewer 1 Report
Comments and Suggestions for Authors
Comments:
It was not possible to evaluate this study because many results are related to a Supplementary Material that was not uploaded.
Comments
Fig. 1 – Please add y-axis (heat flow), indicating the exo or endo process with an arrow.
On L152-157 – It was not clear why soaking for 3h in CaCl2 solution has the same crystallization behavior as the blank (without alginate). Please comment on this.
On L164-170 the authors commented on Figures S2 and S3. However, there is no Supplementary Material related to this submission. It is not possible to understand the text without the images.
To better understand the results, it would be helpful to have the bijel composition. I would like to know how much of oil phase and how much of aqueous phase. Reading the section “4.3. Bijel-like structure synthesis” it was not possible to get it.
On L180-184 – “The system is observable to achieve the equilibrium after the first few minutes, and as it can be deduced, the 20 mg/mL samples showed a higher equilibrium swelling ratio (Figure S4). “ Please provide Figure S4
“Furthermore, comparing it with the blank samples it can be observed that the equilibrium swelling ratio is one order of magnitude higher.” Is the blank the sample without alginate? Please explain it in the text.
“ It is so evident that adding a gelling agent in the hydrophilic domains of biphasic porous structures increases the ability to retain water within the 3D network.” How much alginate was added to the sample analyzed in Figure 3?
Figure 4 – Please inform the temperature in the caption.
Regarding the rheology, it was not possible to evaluate since many results are supposedly in a Supplementary material not shown.
Why fluorescein and rhodaminB? They do not represent models for drug release.
The size of symbols, numbers and lettering in the graphs are very small. If possible, please increase them.
Reviewer 2 Report
Comments and Suggestions for Authors
Title: Biphasic porous bijel-like structures with hydrogel domains as controlled drug delivery systems
In this manuscript, biphasic bijel was preparated and regulated by adding the alginate for drug delivery. And, the effect of alginate addition has been analyzed. It may be interesting to the readers. However, the description of the study was not clear, and the authors did not have a clear understanding of bijel and Pickering emulsions, as well as a clear understanding of hydrogel and aerogel. Therefore, it don't recommend publication in the current version. Some comments for this manuscript before possible acceptance.
1. The abstract should be reorganized due to unambiguous information is not mentioned.
2. The author claimed that “Bijels are a peculiar type of Pickering emulsions”, It's puzzling. Pickering emulsions are systems composed of two immiscible fluids stabilized by nanoparticles. Please explain.
3. Where is the Figure S?
4. In the Figure 1, the melting temperature seems to be the water/ice.
5. What information was provided by Figure 2? If already reported in previous works from author's research group, please removed.
6. L200. Generally, the abbreviation for linear viscoelastic region is LVR, not LVER. E.g, https://doi.org/10.1016/j.carbpol.2022.120499,
7. L205. The full name MAOS and LAOS of should be given.
8. Texture should be considered more than rheology to analyze the properties of the gel.
9. In the system, why the double nature for loading and releasing both hydrophilic and hydrophobic drugs? There are no hydrophobic chambers in the system.
10. L317, where is the pore in the system.
Comments on the Quality of English LanguageExtensive editing of English language required.
Round 2
Reviewer 1 Report
Comments and Suggestions for Authors
The revised manuscript and Supplementary Material are adequate. The only point that was not improved is the quality of the graphs. The size of the numbers and lettering are still small. It could be enlarged to improve the readability.
Reviewer 2 Report
Comments and Suggestions for Authors
Thank you for taking the time to address all of my comments. I am satisfied with the revisions made to the manuscript, but detailed results should be described In the Abstract, not just verbal descriptions.
